# $\mathbb{Z}_2$ topological invariant in three-dimensional PT- and PC-symmetric class CI band structures

Ken Shiozaki

Center for Gravitational Physics and Quantum Information, Yukawa Institute for Theoretical Physics, Kyoto University, Kyoto 606-8502, Japan

September 30, 2025

## Abstract

We construct a previously missing $\mathbb{Z}_2$ topological invariant for three-dimensional band structures in symmetry class CI defined by parity-time (PT) and parity-particle-hole (PC) symmetries. PT symmetry allows one to define a real Berry connection and, based on the $\eta$-invariant, a spin-Chern–Simons (spin-CS) action. We show that PC symmetry quantizes the spin-CS action to $\{0, 2\pi\}$ with $4\pi$ periodicity, thereby yielding a well-defined $\mathbb{Z}_2$ invariant. This invariant is additive under direct sums of isolated band structures, reduces to a known $\mathbb{Z}_2$ index when a global Takagi factorization exists, and in general depends on the choice of spin structure. Finally, we demonstrate lattice models in which this newly introduced $\mathbb{Z}_2$ invariant distinguishes topological phases that cannot be detected by the previously known topological indices.

## 1 Introduction

A central problem in topological band theory involves classifying topological invariants under given symmetries, deriving explicit formulas, and developing practical methods for their computation. In particular, invariants in spatial dimensions three or lower are most relevant to material physics. For the ten Altland–Zirnbauer (AZ) symmetry classes defined by internal symmetries such as time-reversal and particle–hole symmetry [1], the classification and construction of topological invariants is well understood [2–4]. By contrast, for general crystalline symmetries a complete classification is still lacking.

Among crystalline symmetries, those combining spatial inversion with time-reversal or particle-hole conjugation, referred to as parity-time (PT) and parity-particle-hole (PC) symmetries, play an important role because they enforce selection rules on physical response coefficients such as electromagnetic ones. A distinctive feature of PT and PC is that they leave each Bloch momentum $\boldsymbol{k}$ invariant. Since topological invariants are constructed from the local band topology in momentum space, it is of particular importance to clarify how such invariants are formulated under PT and PC symmetries alone. In analogy with onsite symmetries, PT and PC lead to eight real AZ classes. For these eight classes in dimensions three or lower, all topological invariants are known except for the $\mathbb{Z}_2$ index of three-dimensional class CI.

On the other hand, the classification group (such as $\mathbb{Z}$ or $\mathbb{Z}_2$) of topological insulators and superconductors protected by PT and PC symmetries can be obtained from $K$-theory over spheres, and the classification results are already known [5–8]. Moreover, when the parameter space is the three-sphere rather than the three-torus, a $\mathbb{Z}_2$ index for class CI can be defined by the Takagi factorization [9]. It is precisely the case of the three-torus that makes the present problem subtle.

The purpose of this paper is to construct the previously unknown $\mathbb{Z}_2$ invariant of three-dimensional class CI. In brief, the new invariant is given by a spin-Chern–Simons (spin-CS) action with $4\pi$ periodicity induced by PT symmetry [10], which is further quantized to $\{0, 2\pi\}$ by PC symmetry. Since the definition of the spin-CS action requires the $\eta$-invariant [11], we provide the necessary details in this paper. We also demonstrate that the invariant can take nontrivial values in concrete models.

In this paper, we do not address physical phenomena such as corner states, which have already been discussed in [9].

The structure of this paper is as follows. In Sec. 2 we summarize the setting and basic ingredients. In Sec. 3 we present the construction of the new $\mathbb{Z}_2$ invariant and discuss its properties. In Sec. 4 we apply it to explicit models and confirm its nontriviality. Finally, in Sec. 5 we give our conclusion. For completeness, Appendix B collects the invariants of the other real AZ classes defined by PT and PC symmetries in dimensions up to three.

## 2    Preliminaries

We consider a Hamiltonian $H_{\boldsymbol{k}}$ periodically defined on the three-dimensional torus (Brillouin zone, BZ), $\boldsymbol{k} \in T^3$. For each $\boldsymbol{k} \in T^3$, $H_{\boldsymbol{k}}$ is a Hermitian $2n \times 2n$ matrix, and we assume a spectral gap condition that $H_{\boldsymbol{k}}$ has no zero eigenvalue at any $\boldsymbol{k}$. Since our interest lies in topological invariants independent of the specific band dispersion, we flatten the spectrum so that the eigenvalues become $\pm 1$, i.e. $(H_{\boldsymbol{k}})^2 = \mathbf{1}$[1].

At each momenta we impose the PT and PC symmetries that define class CI:

$$TH_{\boldsymbol{k}}^* T^\dagger = H_{\boldsymbol{k}}, \quad TT^* = 1, \tag{1}$$

$$CH_{\boldsymbol{k}}^* C^\dagger = -H_{\boldsymbol{k}}, \quad CC^* = -1, \tag{2}$$

where $M^*$ denotes the complex conjugate of the matrix $M$. Without loss of generality, we may take

$$T = \sigma_x, \quad C = \sigma_y. \tag{3}$$

The combined chiral symmetry $\sigma_z H_{\boldsymbol{k}} \sigma_z = -H_{\boldsymbol{k}}$ implies that the Hamiltonian can be written in the form

$$H_{\boldsymbol{k}} = \begin{pmatrix} & q_{\boldsymbol{k}} \\ q_{\boldsymbol{k}}^\dagger & \end{pmatrix}, \quad q_{\boldsymbol{k}} \in U(n). \tag{4}$$

The PT symmetry condition is equivalent to

$$q_{\boldsymbol{k}}^\top = q_{\boldsymbol{k}}. \tag{5}$$

Thus, the problem is to construct topological invariants that characterize families of symmetric unitary matrices $q_{\boldsymbol{k}}$ parametrized by $T^3$.

In general, the symmetry matrices $T$ and $C$ may depend on momentum $\boldsymbol{k}$, reflecting the Wyckoff positions of the underlying degrees of freedom on which $H_{\boldsymbol{k}}$ is defined. In this paper we restrict to the momentum-independent case, corresponding to all degrees of freedom localized at a common Wyckoff position. This does not reduce generality, since relative Wyckoff displacements can be represented as occupied states of a Hamiltonian defined at a single position.

### 2.1    Real bundle and real Berry connection

Let $\{U_\alpha\}_\alpha$ be a good covering of $T^3$. Locally, a symmetric unitary matrix admits the Takagi factorization

$$q_{\boldsymbol{k}} = Q_{\boldsymbol{k}}^\alpha (Q_{\boldsymbol{k}}^\alpha)^\top, \quad Q_{\boldsymbol{k}}^\alpha \in U(n), \quad \boldsymbol{k} \in U_\alpha. \tag{6}$$

Here $Q_{\boldsymbol{k}}^\alpha$ has an $O(n)$ gauge ambiguity

$$Q_{\boldsymbol{k}}^\alpha \mapsto Q_{\boldsymbol{k}}^\alpha W_{\boldsymbol{k}}^\alpha, \quad W_{\boldsymbol{k}}^\alpha \in O(n). \tag{7}$$

Thus a symmetric unitary takes values in the classifying space $U(n)/O(n)$, and the patch transition is given by $O(n)$ matrices,

$$Q_{\boldsymbol{k}}^\beta = Q_{\boldsymbol{k}}^\alpha S_{\boldsymbol{k}}^{\alpha\beta}, \quad S_{\boldsymbol{k}}^{\alpha\beta} \in O(n), \quad \boldsymbol{k} \in U_\alpha \cap U_\beta. \tag{8}$$

---

[1]For the spectral decomposition $H_{\boldsymbol{k}} = \sum_j E_{j\boldsymbol{k}} \, u_{j\boldsymbol{k}} u_{j\boldsymbol{k}}^\dagger$, the flattening is given by $\operatorname{sgn} H_{\boldsymbol{k}} := \sum_j \operatorname{sgn}(E_{j\boldsymbol{k}}) \, u_{j\boldsymbol{k}} u_{j\boldsymbol{k}}^\dagger$.

On triple overlaps, the two decompositions $Q_{\boldsymbol{k}}^\gamma = Q_{\boldsymbol{k}}^\alpha S_{\boldsymbol{k}}^{\alpha\gamma} = Q_{\boldsymbol{k}}^\beta S_{\boldsymbol{k}}^{\beta\gamma}$ imply the cocycle condition

$$S_{\boldsymbol{k}}^{\alpha\beta} S_{\boldsymbol{k}}^{\beta\gamma} = S_{\boldsymbol{k}}^{\alpha\gamma}, \quad \boldsymbol{k} \in U_\alpha \cap U_\beta \cap U_\gamma. \tag{9}$$

Hence a family $q_{\boldsymbol{k}}$ on $T^3$ defines a rank-$n$ principal $O(n)$ bundle, which we denote it as $P_q$, over $T^3$ and hence Stiefel–Whitney (SW) classes

$$w_i(P_q) \in H^i(T^3; \mathbb{Z}/2). \tag{10}$$

By dimension, $w_i(P_q) = 0$ for $i \geq 4$, and on $T^3$ one has $w_3 = w_1 w_2$ [2]. Thus, the independent $\mathbb{Z}_2$ invariants on $T^3$ are given by the three first and three second SW classes.

In fact, the principal $O(n)$ bundle is introduced from PT symmetry alone. Let $\Phi_{\boldsymbol{k}}^\alpha$ be a local frame of occupied states of the (flattened) Hamiltonian: $H_{\boldsymbol{k}} \Phi_{\boldsymbol{k}}^\alpha = -\Phi_{\boldsymbol{k}}^\alpha, \boldsymbol{k} \in U_\alpha$, where $\Phi_{\boldsymbol{k}}^\alpha \in \mathrm{Mat}_{2n \times n}(\mathbb{C})$ and $(\Phi_{\boldsymbol{k}}^\alpha)^\dagger \Phi_{\boldsymbol{k}}^\alpha = \mathbf{1}_n$. One can impose the real condition

$$T(\Phi_{\boldsymbol{k}}^\alpha)^* = \Phi_{\boldsymbol{k}}^\alpha, \quad \boldsymbol{k} \in U_\alpha \tag{11}$$

locally, and patch transformations preserving (11) are written as $\Phi_{\boldsymbol{k}}^\beta \mapsto \Phi_{\boldsymbol{k}}^\alpha S_{\boldsymbol{k}}^{\alpha\beta}$ with $S_{\boldsymbol{k}}^{\alpha\beta} \in O(n)$, which define a real $O(n)$ bundle. Correspondingly, the $o(n)$ Berry connection is given locally by $A_{\boldsymbol{k}}^\alpha = \Phi_{\boldsymbol{k}}^\alpha d\Phi_{\boldsymbol{k}}^\alpha$ for $\boldsymbol{k} \in U_\alpha$, which satisfies $(A_{\boldsymbol{k}}^\alpha)^\top = -A_{\boldsymbol{k}}^\alpha$ and transforms as $A_{\boldsymbol{k}}^\beta = S_{\boldsymbol{k}}^{\alpha\beta}(A_{\boldsymbol{k}}^\alpha + d)S_{\boldsymbol{k}}^{\alpha\beta}$ over the patch intersection $\boldsymbol{k} \in U_\alpha \cap U_\beta$. With the choice (3), the Takagi factorization (6) yields a local frame satisfying (11):

$$\Phi_{\boldsymbol{k}}^\alpha = \frac{1}{\sqrt{2}} \begin{pmatrix} iQ_{\boldsymbol{k}}^\alpha \\ -i(Q_{\boldsymbol{k}}^\alpha)^* \end{pmatrix}. \tag{12}$$

In this representation, the real Berry connection becomes

$$A_{\boldsymbol{k}}^\alpha = \frac{1}{2}\Big(Q_{\boldsymbol{k}}^\alpha dQ_{\boldsymbol{k}}^\alpha + (Q_{\boldsymbol{k}}^\alpha dQ_{\boldsymbol{k}}^\alpha)^*\Big). \tag{13}$$

On the one hand, without the real condition (11), a global frame

$$\Phi_{\boldsymbol{k}} = \frac{1}{\sqrt{2}} \begin{pmatrix} q_{\boldsymbol{k}} \\ -1_n \end{pmatrix}, \quad \boldsymbol{k} \in T^3 \tag{14}$$

exists, together with the global $u(n)$ Berry connection

$$A_{\boldsymbol{k}} = \frac{1}{2}q_{\boldsymbol{k}}^\dagger dq_{\boldsymbol{k}}. \tag{15}$$

## 2.2 $K$-groups and known topological invariants

The $K$-theory classification of gapped Hamiltonians with symmetries (1), (2) is given by the degree-1 $KO$-group [4–7], which decomposes into contributions from spheres:

$$\begin{aligned} KO^1(T^3) &\cong \widetilde{KO}^1(S^3) \oplus \widetilde{KO}^1(S^2)^{\oplus 3} \oplus \widetilde{KO}^1(S^1)^{\oplus 3} \oplus KO^1(\{\mathrm{pt}\}) \\ &\cong \mathbb{Z}_2 \oplus \mathbb{Z}_2^{\oplus 3} \oplus \mathbb{Z}^{\oplus 3} \oplus 0. \end{aligned} \tag{16}$$

Thus, there must be three independent $\mathbb{Z}$ invariants from the 1-cycles, three $\mathbb{Z}_2$ invariants from the 2-cycles, and one $\mathbb{Z}_2$ invariant from the whole 3-torus.

Let $S_\mu^1$ ($\mu \in \{x, y, z\}$) denote the loops in the $k_\mu$ direction, and $T_{\mu\nu}^2$ ($\mu\nu \in \{xy, yz, zx\}$) the subtori in the $k_\mu k_\nu$ planes. The three $\mathbb{Z}$ invariants are the one-dimensional winding numbers

$$W_{1\mu}[q] = \frac{1}{2\pi i} \oint_{S_\mu^1} d\log \det q_{\boldsymbol{k}} \in \mathbb{Z}, \quad \mu = x, y, z. \tag{17}$$

---

[2] This follows from Wu's formula $Sq^1(w_2) = w_1 w_2 + w_3$, together with the Cartan formula $Sq^1(ab) = Sq^1(a)b + aSq^1(b)$ and the relation $Sq^1(x) = x^2$ for degree-one $x$. For example, in $H^*(T^3; \mathbb{Z}/2) = \mathbb{Z}/2[x, y, z]/(x^2, y^2, z^2)$, we have $Sq^1(xy) = x^2 y + xy^2 = 0$.

Since $\det(qq') = \det q \det q'$, they satisfy the sum rule

$$W_{1\mu}[q \oplus q'] = W_{1\mu}[q] + W_{1\mu}[q']. \tag{18}$$

They are related to the first SW class as[3]:

$$W_{1\mu}[q] \equiv \int_{S^1_\mu} w_1(P_q) \in \{0, 1\}. \tag{19}$$

The $\mathbb{Z}_2$ invariants on 2-cycles are the second SW number:

$$\int_{T^2_{\mu\nu}} w_2(P_q) \in \{0, 1\}. \tag{20}$$

The second SW class $w_2$ does not satisfy a simple additive rule; instead, it obeys the Whitney sum formula

$$w_2(P_{q \oplus q'}) = w_2(P_q) + w_2(P_{q'}) + w_1(P_q)w_1(P_{q'}). \tag{21}$$

See Ref. [12] for the definition of the second SW class and its applications in band theory and topological phases. For efficient computational methods, see also Ref. [13].

For the remaining $\mathbb{Z}_2$ invariant corresponding to $\widetilde{KO}^1(S^3) = \mathbb{Z}_2$, no general expression was known. As a special case, if a global Takagi factorization $q_{\boldsymbol{k}} = Q_{\boldsymbol{k}} Q_{\boldsymbol{k}}^\top$ exists on $T^3$, one has [9]

$$W_3[Q] \mod 2, \tag{22}$$

where $W_3$ is the three-dimensional winding number (see (101) for the definition). Under a global gauge transformation $Q \mapsto QS$, $S : T^3 \to O(n)$, it changes as $W_3[QS] = W_3[Q] + W_3[S]$ with $W_3[S] \in 2\mathbb{Z}$, so only the parity is gauge-invariant.

Note that the winding number of $q_{\boldsymbol{k}}$ itself vanishes due to the symmetry $q_{\boldsymbol{k}}^\top = q_{\boldsymbol{k}}$: $W_3[q_{\boldsymbol{k}}] = 0$. In general, the CS action (or magnetoelectric polarization) $\mathrm{CS}(A)$ of the Berry connection $A$ of occupied states, which takes values in $\mathbb{R}/2\pi\mathbb{Z}$, is well defined for any three-dimensional insulator. For class CI, however, it vanishes,

$$\mathrm{CS}_3(A) = 0 \mod 2\pi, \tag{23}$$

as a consequence of the relation $\mathrm{CS}_3(A) \equiv \pi W_3[q_{\boldsymbol{k}}] \mod 2\pi$ [5]. For the definition of $\mathrm{CS}_3(A)$ and the derivation of Eq. (23), see Sec. A.2 for completeness.

In the next section, we construct the previously unexplored $\mathbb{Z}_2$ invariant for generic class CI Hamiltonians $H_{\boldsymbol{k}}$ defined over three-dimensional spin manifolds.

# 3 Construction of $\mathbb{Z}_2$ invariant

In this section, we see that PT symmetry (1) leads to the "spin-CS action" with $4\pi$ periodicity rather than $2\pi$, and that PC symmetry (2) further quantizes the spin-CS action to $\{0, 2\pi\}$, thereby giving a $\mathbb{Z}_2$ value.

## 3.1 Spin-CS action

The spin-CS action [10] is defined in terms of the $\eta$-invariant [11]. Here we only collect the necessary facts without going into details (for a concise overview in a field-theoretic context, see App. B of Ref. [14]).

---

[3]The first SW class equals the Berry phase quantized to $\mathbb{Z}_2$ under PT symmetry. In the global frame (15), the Berry phase is $\frac{1}{2\pi i} \oint_{S^1_\mu} A_{\boldsymbol{k}} \equiv \frac{1}{2} W_{1\mu}[q]$.

Let $(M, g)$ be a closed oriented three-dimensional Riemannian manifold with Euclidean metric $g$. Such a manifold is spin, and we fix a spin structure $\sigma$ on $M$. For a principal $O(n)$ bundle $P \to M$ with $o(n)$ connection $A$, consider the Dirac operator (in a local expression)

$$D_A = \gamma^\mu (\partial_\mu + i\omega_\mu - iA_\mu), \tag{24}$$

where $\gamma^\mu$s are 2 by 2 gamma matrices and $\omega_\mu$ is the spin connection. The operator $iD_A$ is Hermitian, has a real spectrum, and each eigenvalue $\lambda$ is doubly degenerate due to a quaternionic structure [10][4]. Define

$$\eta_{iD_A}(s) = \sum_{\lambda \neq 0} \text{sgn}\,(\lambda)|\lambda|^{-s}, \tag{25}$$

where $\lambda$ runs over eigenvalues of $iD_A$. The spectral quantity $\eta_{iD_A}(s)$ converges for $\Re(s) > 3/2$ and extends analytically to $s = 0$, giving the $\eta$-invariant $\eta_{iD_A}(0)$ [11]. Introduce the $\xi$-invariant (or the reduced $\eta$-invariant)

$$\xi(P, A, g, \sigma) = \frac{\eta_{iD_A}(0) + \dim \ker(iD_A)}{2} \mod 2, \tag{26}$$

which varies smoothly in $g$ and $A$. Similarly, for the Dirac operator without coupling $D_{\text{triv}} = \gamma^\mu(\partial_\mu + i\omega_\mu)$, we denote its $\xi$-invariant by

$$\xi_{\text{triv}}(g, \sigma) = \frac{\eta_{iD_{\text{triv}}}(0) + \dim \ker(iD_{\text{triv}})}{2}. \tag{27}$$

From the spectral definition (25), $\xi$ is additive under direct sums of bundles and connections[5]:

$$\xi(P \oplus P', A \oplus A', g, \sigma) = \xi(P, A, g, \sigma) + \xi(P', A', g, \sigma). \tag{28}$$

It follows from the APS index theorem that the difference of $\xi$-invariants for Dirac operators coupled to two real representations $\rho_1$ and $\rho_2$ of $O(n)$ with the same dimension $\dim \rho_1 = \dim \rho_2$ is invariant under smooth deformations of the metric $g$ [10]. Since $n\xi_{\text{triv}}(g, \sigma)$ corresponds to $n$ copies of the trivial representation, the combination $\xi(P, A, g, \sigma) - n\xi_{\text{triv}}(g, \sigma)$ is independent of $g$ and thereby defines the spin-CS action

$$\text{CS}_{\text{spin}}(P, A, \sigma) := 2\pi \left( \xi(P, A, g, \sigma) - n\xi_{\text{triv}}(g, \sigma) \right) \mod 4\pi. \tag{29}$$

From (28), it also satisfies the sum rule

$$\text{CS}_{\text{spin}}(P \oplus P', A \oplus A', \sigma) \equiv \text{CS}_{\text{spin}}(P, A, \sigma) + \text{CS}_{\text{spin}}(P', A', \sigma) \mod 4\pi. \tag{30}$$

See [10, 15] and references therein for further properties and applications in field theory. Here we keep the principal $O(n)$ bundle $P$ in the argument of $\text{CS}_{\text{spin}}$, since the local patch connection $A^\alpha$ does not capture the flat bundle data that can modify the spin-CS action (as well as the ordinary CS action). The following lemma shows that $\text{CS}_{\text{spin}}(P, A, \sigma)$ generalizes the ordinary CS action [16].

---

**Lemma 3.1.** *If $w_1(P) = 0$, one can choose a four-dimensional spin manifold $X$ with boundary $\partial X = M$ together with an extension $\tilde{A}$ of the $so(n)$ connection $A$ from $M$ to $X$. In this case, the spin-CS action is written as*

$$\text{CS}_{\text{spin}}(P, A, \sigma) \equiv -\frac{1}{4\pi} \int_X \text{tr}\,[\tilde{F}^2] \mod 4\pi, \tag{31}$$

*where $\tilde{F} = d\tilde{A} + \tilde{A}^2$ is the curvature. This expression is independent of the particular choice of $X$ and $\tilde{A}$. Moreover, reducing modulo $2\pi$ reproduces the ordinary CS action of a $u(n)$ connection:*

$$\text{CS}_{\text{spin}}(P, A, \sigma) \equiv \text{CS}(A) \mod 2\pi. \tag{32}$$

---

[4]Locally, on flat Euclidean space, the Dirac operator is $iD_A = \sum_{\mu=x,y,z}(i\partial_{k_\mu} + A_{k_\mu})\sigma_\mu$, which has the symmetry $\sigma_y(iD_A)^*\sigma_y = iD_A$. This leads to Kramers degeneracy.
[5]The direct sum of connections $A \oplus A'$ means that the off-diagonal elements between $P$ and $P'$ are absent.

*Proof.* If $w_1(P) = 0$, the bundle $P$ is a principal $SO(n)$ bundle, classified by maps $[M, BSO(n)]$. In fact, the spin cobordism group in three dimensions vanishes $\Omega_3^{\text{spin}}(BSO(n)) = 0$ [6]. Hence, whenever $w_1(P) = 0$, there exists a bounding four-spin manifold $X$ with $\partial X = M$, together with an $so(n)$ connection $\tilde{A}$ on $X$ that restricts to $A$ on the boundary: $\tilde{A}|_M = A$.

Let $\tilde{D}_{\tilde{A}}$ denote the Dirac operator on $X$ coupled with the $so(n)$ connection $\tilde{A}$, and $\tilde{D}_{\text{triv}}$ the uncoupled Dirac operator. By the APS index theorem [11], one finds

$$\text{Ind}(i\tilde{D}_{\tilde{A}}) = n \int_X \frac{1}{196\pi^2} \text{tr}\,[\tilde{R}^2] - \int_X \frac{1}{8\pi^2} \text{tr}\,[\tilde{F}^2] - \xi(P, A, g, \sigma), \tag{33}$$

$$\text{Ind}(i\tilde{D}_{\text{triv}}) = \int_X \frac{1}{196\pi^2} \text{tr}\,[\tilde{R}^2] - \xi(g, \sigma). \tag{34}$$

Because of the quaternionic structure, the index on the left-hand side is always even. This leads to the relation (31). The possible ambiguity from choosing different extensions $(X, \tilde{A})$ and $(X', \tilde{A}')$ is given by the integral of the Pontryagin class on the closed manifold $X'' = X \cup (-X')$. Standard arguments [16] using the Wu formula show that this ambiguity is always a multiple of $4\pi$ [7]. Therefore, the expression (31) is well defined modulo $4\pi$. Finally, embedding $O(n) \hookrightarrow U(n)$ gives (32). $\qquad\square$

Note that in the right-hand side of (31), the dependence on the spin structure is hidden in the choice of the four-dimensional extension $X$.

When $w_1(P) \neq 0$, the spin-CS action $\text{CS}_{\text{spin}}(P, A, \sigma)$ cannot be written using a four-dimensional extension. Nevertheless, the definition via the $\eta$-invariant (29) is still well-defined.

## 3.2 $\mathbb{Z}_2$ invariant

We now construct a $\mathbb{Z}_2$ number for a symmetric unitary matrix $q$.

---

**Definition 3.2.** For a family of symmetric unitary matrices on a three-dimensional spin manifold $M$, $q : M \to U(n)$ with $q^\top = q$, let $P_q$ be the principal $O(n)$ bundle defined by the local Takagi factorization (6), and let $A_q$ be the $o(n)$ Berry connection defined by (13). Fix a spin structure $\sigma$ on $M$. We define the mod-2 valued $\nu$-invariant as the spin-CS action

$$\nu[q, \sigma] := \frac{1}{2\pi} \text{CS}_{\text{spin}}(P_q, A_q, \sigma) \mod 2. \tag{35}$$

---

We show that $\nu[q, \sigma]$ is the desired $\mathbb{Z}_2$ number.

---

**Proposition 3.3.** *(i) Let $q_i : M \to U(n_i)$ with $q_i^\top = q_i$ for $i = 0, 1$ be two families of symmetric unitaries. Then the sum rule holds*

$$\nu[q_0 \oplus q_1, \sigma] = \nu[q_0, \sigma] + \nu[q_1, \sigma]. \tag{36}$$

*(ii) If $w_1(P_q) = 0$, then $\nu[q, \sigma]$ is quantized to $\{0, 1\}$:*

$$\nu[q, \sigma] \in \{0, 1\}. \tag{37}$$

*(iii) If the principal $O(1)$ bundle $\det P_q \in H^1(M, \mathbb{Z}_2)$ is the mod 2 reduction of an element of $H^1(M, \mathbb{Z})$, then $\nu[q, \sigma]$ is quantized to $\{0, 1\}$. In particular, for $M = T^3$ this is always the case.*

---

[6]See Appendix B.4 of [17] for a computation using the Atiyah–Hirzebruch spectral sequence. On the $E^2$ page, $E_{2,1}^2 = H_2(BSO(n \geq 2), \Omega_1^{\text{spin}}) = \mathbb{Z}_2$ remains. On the other hand, $E_{4,0}^2 = H_4(BSO(n \geq 2), \mathbb{Z}) = \mathbb{Z}$, and the differential $d_{4,0}^2 : E_{4,0}^2 \to E_{2,1}^2$ is given by the composition of the mod 2 reduction and the dual of $Sq^2$. Since $p_1 \equiv w_2^2$ mod 2 and $Sq^2 w_2 = w_2^2$, the Pontryagin number $\int_X p_1$ of an $SO$ bundle on a four-dimensional spin manifold $X$ is always even, corresponding to $\ker d_{4,0}^2$.

[7]$\int_{X''} \frac{1}{8\pi^2} \text{tr}\,[\tilde{F}^2] = \langle p_1(P), [X''] \rangle \equiv \langle w_2(P)^2, [X''] \rangle$ mod 2. Using the Wu formula for $w_1(P) = 0$, $\langle w_2(P)^2, [X''] \rangle = \langle w_2(TX'') \cup w_2(P), [X''] \rangle$, and since $X''$ is spin, $w_2(TX'') = 0$, thus the difference is always even.

*Proof.* (i) Follows from the additivity (28) of the $\xi$-invariant. (ii) It suffices to show $\mathrm{CS}(A_q) \equiv 0 \bmod 2\pi$, which is already shown in (23).

(iii) For a $\mathbb{Z}/2$ bundle $\ell \in H^1(M, \mathbb{Z}/2)$, the principal $O(1)$ bundle $(-1)^\ell$ is defined and the tensor product $P \otimes (-1)^\ell$ corresponds to replacing the patch transitions by $S^{\alpha\beta} \mapsto S^{\alpha\beta}(-1)^{\ell_{\alpha\beta}}$, which is equivalent to the shift of spin structure $\sigma \mapsto \sigma + \ell$. Since multiplying patch transitions by constant factors does not change the local $o(n)$ connections $A_\alpha$, we obtain

$$\xi(P \otimes (-1)^\ell, A, g, \sigma) = \xi(P, A, g, \sigma + \ell). \tag{38}$$

Thus, for the spin-CS action we have

$$\frac{1}{2\pi}\mathrm{CS}_{\mathrm{spin}}(P \otimes (-1)^\ell, A, \sigma) = \xi(P \otimes (-1)^\ell, A, g, \sigma) - n\xi_{\mathrm{triv}}(g, \sigma)$$
$$= \xi(P, A, g, \sigma + \ell) - n\xi_{\mathrm{triv}}(g, \sigma)$$
$$= \frac{1}{2\pi}\mathrm{CS}_{\mathrm{spin}}(P, A, \sigma) + n\,\Delta\xi_{\mathrm{triv}}(\sigma; \ell) \quad \bmod 2, \tag{39}$$

where we introduced the notation

$$\Delta\xi_{\mathrm{triv}}(\sigma; \ell) := \xi_{\mathrm{triv}}(g, \sigma + \ell) - \xi_{\mathrm{triv}}(g, \sigma). \tag{40}$$

The quantity $\Delta\xi_{\mathrm{triv}}(\sigma; \ell)$ is known as the $\mathbb{Z}_8$ invariant of (2+1)-dimensional invertible fermionic phases with an onsite $\mathbb{Z}_2$ symmetry [18, 19] and can take quarter-integer values in general [15, 20] [8].

If $n$ is odd, $\det(-\mathbf{1}_n) = -1$, and for the $O(n)$ patch transitions $S^{\alpha\beta}(-1)^{\ell_{\alpha\beta}}$, we have $\det[S^{\alpha\beta}(-1)^{\ell_{\alpha\beta}}] = (\det S^{\alpha\beta}) \cdot (-1)^{\ell_{\alpha\beta}}$, so by choosing $(-1)^\ell$ to be the determinant line bundle $\det P$ one can always deform to $w_1 = 0$:

$$w_1(P \otimes \det P) = 0, \quad w_2(P \otimes \det P) = w_2(P), \quad (n \text{ odd}). \tag{41}$$

Writing $\det P = (-1)^{\ell_P}$, and using the already established case $w_1(P_q) = 0$ where $\nu[q, \sigma] \in \{0, 1\}$, we obtain

$$\nu[q, \sigma] = \frac{1}{2\pi}\mathrm{CS}_{\mathrm{spin}}(P_q \otimes (-1)^{\ell_{P_q}}, A_q, \sigma + \ell_{P_q}) - n\,\Delta\xi_{\mathrm{triv}}(\sigma; \ell_{P_q})$$
$$\underset{\bmod 1}{\equiv} -n\,\Delta\xi_{\mathrm{triv}}(\sigma; \ell_{P_q}). \tag{42}$$

For the spin-structure dependence of the $\xi$-invariant, the following was established: If $A$ is flat and $\ell$ is the mod 2 reduction of an element of $H^1(M, \mathbb{Z})$, then $\xi(P, A_{\mathrm{flat}}, g, \sigma + \ell) - \xi(P, A_{\mathrm{flat}}, g, \sigma) \in \mathbb{Z}$ (Corollary 3.2. in [20]). Since we are in the trivially coupled case here, the claim follows. In particular, for $M = T^3$ we have $H^1(T^3, \mathbb{Z}/2) = \mathbb{Z}_2^{\times 3}$ obtained as the mod 2 reduction of generators of $H^1(T^3, \mathbb{Z}) = \mathbb{Z}^{\times 3}$. The fact that $\Delta\xi_{\mathrm{triv}}(\sigma, \ell) \in \mathbb{Z}$ for $M = T^3$ is also confirmed from the explicit calculation of the $\xi$-invariant. (See (86) below.)

If $n$ is even, add the trivial $O(1)$ bundle $\underline{\mathbb{R}}$ with the trivial connection "0" and use the sum rule (30), reducing to the odd-$n$ case. $\qquad\square$

Thus we have shown that $\nu[q, \sigma]$ indeed behaves as the desired $\mathbb{Z}_2$ invariant. Note that $\nu[q, \sigma]$ depends explicitly on the spin structure $\sigma$. To see that $\nu[q, \sigma]$ can actually take a nontrivial value, we show below that in the case where $q$ admits a global Takagi factorization $q = QQ^\top$, the $\nu$-invariant reduces to the known expression (22) for the $\mathbb{Z}_2$ invariant.

**Lemma 3.4.** *Let $h : M \to U(n)$ be a smooth map. Then*

$$\nu[hqh^\top, \sigma] - \nu[q, \sigma] \equiv W_3[h] \quad \bmod 2. \tag{43}$$

---

[8]The $\mathbb{Z}_8$–valued invariant $\Delta\xi_{\mathrm{triv}}(\sigma; \ell)$ admits the following explicit expression [21, 22]: let $\mathrm{PD}(\ell)$ denote the Poincaré dual surface to $\ell \in H^1(M, \mathbb{Z}_2)$. A spin structure $\sigma$ on $M$ canonically induces a $\mathrm{pin}_-$ structure $\sigma|_{\mathrm{PD}(\ell)}$ on $\mathrm{PD}(\ell)$. Then the $\mathbb{Z}_8$ invariant is given by the Arf–Brown–Kervaire invariant of the quadratic enhancement corresponding to the $\mathrm{pin}_-$ structure. In the special case $M = T^3$, the surface $\mathrm{PD}(\ell)$ is oriented (indeed a two-torus), so the induced $\mathrm{pin}_-$ structure reduces to a spin structure and it reduces to the $\mathbb{Z}_2$ Arf invariant of spin manifolds.

*Proof.* For two $o(n)$ connections $A, A'$ on a principal $O(n)$ bundle $P$, consider the interpolation

$$A_t = (1 - t)A + tA'. \tag{44}$$

Applying the APS index theorem to the Dirac operator over $M \times [0, 1]$, we have

$$\text{Ind}(i\tilde{D}_{A_t}) \equiv -\frac{1}{8\pi^2} \int_{M \times [0,1]} \text{tr}\,[F_t^2] - \xi(P, A, g, \sigma) + \xi(P, A', g, \sigma) \tag{45}$$

$$\equiv -\frac{1}{8\pi^2} \int_{M \times [0,1]} \text{tr}\,[F_t^2] - \frac{1}{2\pi}\text{CS}_{\text{spin}}(P, A', \sigma) + \frac{1}{2\pi}\text{CS}_{\text{spin}}(P, A, \sigma) \quad \text{mod 2.} \tag{46}$$

Since the index on the left-hand side is even,

$$\frac{1}{2\pi}\text{CS}_{\text{spin}}(P, A', \sigma) - \frac{1}{2\pi}\text{CS}_{\text{spin}}(P, A, \sigma) \equiv -\frac{1}{8\pi^2} \int_{M \times [0,1]} \text{tr}\,[F_t^2] \quad \text{mod 2.} \tag{47}$$

Performing the $t$-integration with $A_t = A + tB$, $B = A' - A$, gives representation

$$-\frac{1}{8\pi^2} \int_{M \times [0,1]} \text{tr}\,[F_t^2] = -\frac{1}{8\pi^2} \int_M 2\,\text{tr}\big[(dA + A^2)B + AB^2\big] - \frac{1}{8\pi^2} \int_M \text{tr}\,\big(BdB + \tfrac{2}{3}B^3\big). \tag{48}$$

Here, the integrands are expressed locally, and patch indices are omitted.

Now consider a local Takagi factorization $q = QQ^\top$ of $q$. For the transformed symmetric unitary $hqh^\top$, a local factorization is given by $hqh^\top = (hQ)(hQ)^\top$, so the same patch transition (8) applies. Hence $P_q$ and $P_{hqh^\top}$ define the same principal $O(n)$ bundle, and thus $A_q$ and $A_{hqh^\top}$ can be interpolated linearly. In local form (patch indices suppressed again),

$$A = A_q = \Phi^\dagger d\Phi, \quad \Phi = \frac{1}{\sqrt{2}}\begin{pmatrix} iQ \\ -iQ^* \end{pmatrix}, \tag{49}$$

$$A' = A_{hqh^\top} = \left[\begin{pmatrix} h \\ & h^* \end{pmatrix}\Phi\right]^\dagger d\left[\begin{pmatrix} h \\ & h^* \end{pmatrix}\Phi\right] = A_q + B, \tag{50}$$

$$B = \Phi^\dagger X \Phi, \quad X = \begin{pmatrix} h^\dagger dh \\ & h^\top dh^* \end{pmatrix}. \tag{51}$$

The integrand can be rewritten using the gauge-invariant orthogonal projection

$$P = \Phi\Phi^\dagger = \frac{1}{2}\begin{pmatrix} 1 & -q \\ -q^\dagger & 1 \end{pmatrix} \tag{52}$$

and the matrix $X$ as in

$$\text{tr}\,[(dA + A^2)B] = \text{tr}\,[PdPdPX], \tag{53}$$

$$\text{tr}\,[AB^2] = \text{tr}\,[PdPXPX - \Phi d\Phi^\dagger XPX], \tag{54}$$

$$\text{tr}\,[BdB] = \text{tr}\,[PXPdX + (\Phi d\Phi^\dagger - d\Phi\Phi^\dagger)XPX], \tag{55}$$

$$\text{tr}\,[B^3] = \text{tr}\,[PXPXPX]. \tag{56}$$

Thus

$$\text{tr}\,[F_t^2] = \text{tr}\,\big(2PdPdPX + 2PdPXPX + PXPdX - dPXPX + \tfrac{2}{3}PXPXPX\big). \tag{57}$$

Using $dX = -X^2$, the contributions split into quadratic and cubic terms in $X$. Substituting explicit forms of $P$ and $X$, the quadratic terms become total derivatives

$$\text{tr}\,[2PdPdPX + 2PdPXPX - dPXPX] = \frac{1}{4}d\,\text{tr}\,[qdq^\dagger h^\dagger dh + q^\dagger dqh^\top dh^*] \tag{58}$$

and integrate to zero, while the cubic terms yield

$$\text{tr}\,[PXPdX + \tfrac{2}{3}PXPXPX] = -\tfrac{1}{3}\text{tr}\,[(h^\dagger dh)^3]. \tag{59}$$

Hence,

$$\nu[hqh^\top, \sigma] - \nu[q, \sigma] = -\frac{1}{8\pi^2} \int_{M\times[0,1]} \mathrm{tr}\,[F_t^2] = \frac{1}{24\pi^2} \int_M \mathrm{tr}\,[(h^\dagger dh)^3] = W_3[h]. \quad \square \tag{60}$$

**Corollary 3.5.** *If a symmetric unitary $q : M \to U(n)$ with $q^\top = q$ admits a global Takagi factorization $q = QQ^\top$, then*

$$\nu[q, \sigma] - \nu[\mathbf{1}_n, \sigma] \equiv W_3[Q] \mod 2. \tag{61}$$

## 3.3 Spin structure dependence of $\nu[q, \sigma]$

The constructed $\mathbb{Z}_2$ invariant $\nu[q, \sigma]$ depends explicitly on the spin structure $\sigma$ on $M$. In particular, for the case of interest $M = T^3$, there are $|H^1(T^3, \mathbb{Z}/2)| = 8$ distinct spin structures.

The spin-structure dependence of the spin-CS action was studied in Ref. [10]. Let us introduce the notation

$$\Delta \mathrm{CS}_{\mathrm{spin}}(P, \sigma; \ell) := \mathrm{CS}_{\mathrm{spin}}(P, A, \sigma + \ell) - \mathrm{CS}_{\mathrm{spin}}(P, A, \sigma) \quad \text{for } \ell \in H^1(M, \mathbb{Z}/2). \tag{62}$$

**Proposition 3.6** (Proposition 1.7 in [10]). *For $\ell \in H^1(M, \mathbb{Z}/2)$:*

*(i) If $w_1(P) = w_2(P) = 0$, then the spin-CS action is independent of the spin structure:*

$$\Delta \mathrm{CS}_{\mathrm{spin}}(P, \sigma; \ell) \equiv 0 \mod 4\pi. \tag{63}$$

*(ii) If $w_1(P) = 0$, then*

$$\Delta \mathrm{CS}_{\mathrm{spin}}(P, \sigma; \ell) \equiv 2\pi \int_M w_2(P) \cup \ell \mod 4\pi. \tag{64}$$

*(iii) In general, $\Delta \mathrm{CS}_{\mathrm{spin}}(P, \sigma; \ell)$ is a $\mathbb{Z}/4$ quadratic refinement of the bilinear form*

$$H^1(M, \mathbb{Z}/2) \times H^1(M, \mathbb{Z}/2) \to \mathbb{Z}/2, \quad (\ell_1, \ell_2) \mapsto \int_M w_1(P) \cup \ell_1 \cup \ell_2. \tag{65}$$

This result will be assumed throughout the present paper [9].

We now examine an explicit formula for the $\mathbb{Z}/4$ quadratic refinement in the generic case, including $w_1(P) \neq 0$ or $w_2(P) \neq 0$.

From the additivity of the spin-CS action (30) we have

$$\Delta \mathrm{CS}_{\mathrm{spin}}(P \oplus P', \sigma; \ell) = \Delta \mathrm{CS}_{\mathrm{spin}}(P, \sigma; \ell) + \Delta \mathrm{CS}_{\mathrm{spin}}(P', \sigma; \ell). \tag{66}$$

Since for any principal $O(n)$ bundle $P$ one has $w_1(P^{\oplus 4}) = w_2(P^{\oplus 4}) = 0$, it follows that

$$\Delta \mathrm{CS}_{\mathrm{spin}}(P, \sigma; \ell) \in \{0, \pi, 2\pi, 3\pi\}, \tag{67}$$

and is invariant under smooth deformations of the $o(n)$ connection $A$.

Writing $\det P = (-1)^{\ell_P}$ so that $w_1(P) = \ell_P$, and using Proposition 3.6 (ii), we obtain

$$\Delta \mathrm{CS}_{\mathrm{spin}}(P, \sigma; \ell) = \Delta \mathrm{CS}_{\mathrm{spin}}(P \oplus \det P, \sigma; \ell) - \Delta \mathrm{CS}_{\mathrm{spin}}(\det P, \sigma; \ell)$$

$$= 2\pi \int_M (w_2(P) + w_1(P)^2) \cup \ell - \Delta \mathrm{CS}_{\mathrm{spin}}(\det P, \sigma; \ell). \tag{68}$$

---

[9]Note that the proof in [10] contains some inaccuracies, such as $\Omega_3^{\mathrm{spin}}(BSO(n)) = \mathbb{Z}_2$.

The second term can be written as the second difference

$$\frac{1}{2\pi}\Delta\mathrm{CS}_{\mathrm{spin}}(\det P, \sigma; \ell) = \Delta^2 \xi_{\mathrm{triv}}(\sigma; \ell, \ell_P). \tag{69}$$

Here,

$$\Delta^2 \xi_{\mathrm{triv}}(\sigma; \ell_1, \ell_2) := \Delta\xi_{\mathrm{triv}}(\sigma + \ell_1; \ell_2) - \Delta\xi_{\mathrm{triv}}(\sigma; \ell_2) \tag{70}$$

is symmetric under exchange of $\ell_1, \ell_2$. Thus the change in the spin-CS action under $\sigma \mapsto \sigma + \ell$ is

$$\frac{1}{2\pi}\Delta\mathrm{CS}_{\mathrm{spin}}(P, \sigma; \ell) = \int_M (w_2(P) + w_1(P)^2) \cup \ell - \Delta^2\xi_{\mathrm{triv}}(\sigma; \ell, w_1(P)). \tag{71}$$

The same transformation law holds for the $\nu$-invariant:

$$\nu[q, \sigma + \ell] - \nu[q, \sigma] = \int_M (w_2(P) + w_1(P)^2) \cup \ell - \Delta^2\xi_{\mathrm{triv}}(\sigma; \ell, w_1(P)). \tag{72}$$

Combining with (39), the variation under tensoring with $(-1)^\ell$ is

$$\begin{aligned}
&\frac{1}{2\pi}\Delta\mathrm{CS}_{\mathrm{spin}}(P \otimes (-1)^\ell, A, \sigma) - \frac{1}{2\pi}\Delta\mathrm{CS}_{\mathrm{spin}}(P, A, \sigma) \\
&= \int_M (w_2(P) + w_1(P)^2) \cup \ell + n\Delta\xi_{\mathrm{triv}}(\sigma; \ell) - \Delta^2\xi_{\mathrm{triv}}(\sigma; \ell, w_1(P)).
\end{aligned} \tag{73}$$

We note that even for $M = T^3$, the spin-structure dependence cannot be eliminated.

## 3.4 Alternative formulations of the $\mathbb{Z}_2$ invariant

The invariant $\nu[q, \sigma]$ introduced above is quantized to $\{0, 1\}$ when $M = T^3$, but not necessarily for a general $M$. In this subsection, we sketch alternative definitions obtained by adding or tensoring with $\det P$ so that $w_1$ is removed.

### 3.4.1 Direct sum case

Consider the direct sum with $\det P$, whose $O(1)$ connection is the trivial "0". Define

$$\nu'[q, \sigma] := \frac{1}{2\pi}\mathrm{CS}_{\mathrm{spin}}(P_q \oplus \det P_q, A_q \oplus 0, \sigma) = \nu[q, \sigma] + \Delta\xi_{\mathrm{triv}}(\sigma; w_1(P_q)). \tag{74}$$

Since $w_1(P_q \oplus \det P_q) = 0$, Proposition 3.3(ii) guarantees the quantization

$$\nu'[q, \sigma] \in \{0, 1\} \tag{75}$$

for any three-dimensional spin manifold $M$. On the other hand, the sum rule (36) is modified:

$$\begin{aligned}
\nu'[q \oplus q', \sigma] - \nu'[q, \sigma] - \nu'[q', \sigma] &= \Delta\xi_{\mathrm{triv}}(\sigma; w_1(P_q) + w_1(P_{q'})) - \Delta\xi_{\mathrm{triv}}(\sigma; w_1(P_q)) - \Delta\xi_{\mathrm{triv}}(\sigma; w_1(P_{q'})) \\
&= \Delta^2\xi_{\mathrm{triv}}(\sigma; w_1(P_q), w_1(P_{q'})).
\end{aligned} \tag{76}$$

### 3.4.2 Tensor product case

Let $\mathbb{R}$ denote the trivial $O(1)$ bundle. To eliminate $w_1$, define

$$\begin{aligned}
\nu''[q, \sigma] &:= \frac{1}{2\pi} \times \begin{cases} \mathrm{CS}_{\mathrm{spin}}(P_q \otimes \det P_q, A_q, \sigma) & (n \in \mathrm{odd}), \\ \mathrm{CS}_{\mathrm{spin}}((P_q \oplus \mathbb{R}) \otimes \det P_q, A_q \oplus 0, \sigma) & (n \in \mathrm{even}). \end{cases} \\
&= \nu[q, \sigma] + \int_M (w_2(P_q) + w_1(P_q)^2)\, w_1(P_q) + \begin{cases} (n+2)\Delta\xi_{\mathrm{triv}}(\sigma; w_1(P_q)) & (n \in \mathrm{odd}), \\ (n+3)\Delta\xi_{\mathrm{triv}}(\sigma; w_1(P_q)) & (n \in \mathrm{even}). \end{cases}
\end{aligned} \tag{77}$$

Here we used $\Delta^2 \xi_{\text{triv}}(\sigma; \ell, \ell) = -2\Delta \xi_{\text{triv}}(\sigma; \ell)$. By Proposition 3.3 (ii), this guarantees quantization

$$\nu''[q, \sigma] \in \{0, 1\} \tag{78}$$

for any three-dimensional spin manifold $M$. However, the sum rule is again modified.

To see this, for simplicity, assume that $H^1(M, \mathbb{Z}/2)$ arises as the mod 2 reduction of $H^1(M, \mathbb{Z})$. In this case $\Delta \xi_{\text{triv}}(\sigma; \ell) \in \{0, 1\}$ holds [20]. This condition is satisfied for the case of interest $M = T^3$. Using $\Delta \xi_{\text{triv}}(\sigma; \ell + \ell') = \Delta^2 \xi_{\text{triv}}(\sigma; \ell, \ell') - \Delta \xi_{\text{triv}}(\sigma; \ell) - \Delta \xi_{\text{triv}}(\sigma; \ell')$, we obtain

$$\nu''[q \oplus q', \sigma] - \nu''[q, \sigma] - \nu''[q', \sigma] = \int_M \left( w_2(P_q) w_1(P_{q'}) + w_1(P_q) w_2(P_{q'}) \right) + \Delta^2 \xi_{\text{triv}}(\sigma; \ell, \ell'). \tag{79}$$

Here, the first correction term on the right-hand side of (79) is analogous to the Whitney sum formula for the third SW class,

$$w_3(P \oplus P') = w_3(P) + w_3(P') + w_2(P) w_1(P') + w_1(P) w_2(P'). \tag{80}$$

This is consistent with the fact that the third cohomology group with $\mathbb{Z}_2$ coefficients of $U/O$ is given by $H^3(U/O; \mathbb{Z}/2) = \iota^* w_3$ for the map $\iota : U/O \to BO$ [23].

# 4  Applications

In this section we compute the invariant $\nu[q, \sigma]$ for several models in the case $M = T^3$, which is relevant in band theory. As stated in Proposition 3.3 (iii), in this case the invariant $\nu[q, \sigma]$ is quantized to $\{0, 1\}$.

## 4.1  $1 \times 1$ model

Fix an arbitrary spin structure $\sigma$ on $T^3$. As the simplest model, take a three-dimensional integer vector $\boldsymbol{n} = (n_x, n_y, n_z) \in \mathbb{Z}^{\times 3}$ and consider the $1 \times 1$ model

$$q_{1, \boldsymbol{n}} = e^{i \boldsymbol{n} \cdot \boldsymbol{k}}. \tag{81}$$

For even vectors $2\boldsymbol{m}$, the relation (43) implies

$$\nu[q_{1, \boldsymbol{n} + 2\boldsymbol{m}}, \sigma] = \nu[q_{1, \boldsymbol{n}}, \sigma], \tag{82}$$

so it suffices to restrict to $\boldsymbol{n} \in \{0, 1\}^{\times 3}$. The one-dimensional winding number (17) can be written in vector form as

$$\boldsymbol{W}_1 = \boldsymbol{n}. \tag{83}$$

Hence the first SW class, written in the basis $dk_x, dk_y, dk_z$ of $H^1(T^3, \mathbb{Z}/2) \cong (\mathbb{Z}/2)^{\times 3}$, is

$$w_1(P_{q_{1, \boldsymbol{n}}}) = n_x \, dk_x + n_y \, dk_y + n_z \, dk_z. \tag{84}$$

The second SW class vanishes by dimensional reasons: $w_2(P_{q_{1, \boldsymbol{n}}}) = 0$.

A local Takagi factorization near $\boldsymbol{k} = \boldsymbol{0}$ is given by $e^{i \boldsymbol{n} \cdot \boldsymbol{k}} = Q_{\boldsymbol{k}} Q_{\boldsymbol{k}}^\top$ with $Q_{\boldsymbol{k}} = e^{i \boldsymbol{n} \cdot \boldsymbol{k}/2}$, meaning that a loop $k_\mu \to k_\mu + 2\pi$ in the $\mu$-direction picks up an $O(1)$ phase $(-1)$. Thus the $O(1)$ bundle $P_{q_{1, \boldsymbol{n}}}$ corresponds to a $\mathbb{Z}_2$ twist of the fermionic boundary condition in the $\boldsymbol{n}$-direction. Accordingly, using the $\xi$-invariant one obtains

$$\nu[e^{i \boldsymbol{n} \cdot \boldsymbol{k}}, \sigma] = \xi_{\text{triv}}(g, \sigma + \boldsymbol{n}) - \xi_{\text{triv}}(g, \sigma) = \Delta \xi_{\text{triv}}(\sigma; \boldsymbol{n}). \tag{85}$$

Since the invariant $\nu[e^{i \boldsymbol{n} \cdot \boldsymbol{k}}, \sigma]$ is independent of the metric $g$, we may assume the flat metric $g_{\text{flat}}$ to evaluate the $\xi$-invariant. The spectral function $\eta_{iD_{\text{triv}}}(s)$ does not contribute to the $\xi$-invariant due to the spectral symmetry $\lambda \mapsto -\lambda$ of the Dirac operator $iD_{\text{triv}} = -\boldsymbol{k} \cdot \boldsymbol{\sigma}$. Hence only the zero modes of $iD_{\text{triv}}$

contribute to the $\xi$-invariant, and these occur solely in the RRR sector (periodic boundary conditions for fermions). We have

$$\xi_{\text{triv}}(g_{\text{flat}}, \sigma) \begin{cases} 1 & \text{if } \sigma \text{ is the RRR sector,} \\ 0 & \text{otherwise.} \end{cases} \tag{86}$$

Hence,

$$\nu[q_{1,\boldsymbol{n}}, \sigma] = \begin{cases} 1 & \text{if either } \sigma \text{ or } \sigma + \boldsymbol{n} \text{ is the RRR sector,} \\ 0 & \text{otherwise.} \end{cases} \tag{87}$$

We emphasize that $\nu[q_{1,\boldsymbol{n}}, \sigma]$ depends explicitly on the spin structure $\sigma$.

For direct sums of models of type $q_{1,\boldsymbol{n}}$, the topological numbers $W_1, w_2, \nu$ are completely determined by the sum rules (18), (21), and (36), as demonstrated below.

## 4.2 Example of a model detected only by the $\nu$-invariant

As an example of a model that is first distinguished by the $\nu$-invariant, consider the following direct-sum models:

$$q_A = \begin{pmatrix} e^{ik_x} & & & \\ & e^{ik_y} & & \\ & & e^{ik_z} & \\ & & & e^{-i(k_x+k_y+k_z)} \end{pmatrix}, \tag{88}$$

$$q_B = \begin{pmatrix} 1 & & & \\ & e^{i(k_x-k_y)} & & \\ & & e^{i(k_y-k_z)} & \\ & & & e^{i(k_z-k_x)} \end{pmatrix}. \tag{89}$$

Both models $q_A$ and $q_B$ have trivial one-dimensional winding numbers $W_{1x} = W_{1y} = W_{1z} = 0$. By the sum rule (21), their second SW classes are

$$w_2(P_{q_A}) = w_2(P_{q_B}) = dk_x dk_y + dk_y dk_z + dk_z dk_x. \tag{90}$$

On the other hand, when the spin structure is $\sigma = \sigma_{\text{NS}} := (\text{NS}, \text{NS}, \text{NS})$ (anti-periodic boundary conditions), the $\nu$-invariants of $q_A$ and $q_B$ are given respectively as

$$\begin{aligned} \nu[q_A, \sigma_{\text{NS}}] &= \nu[e^{ik_x}, \sigma_{\text{NS}}] + \nu[e^{ik_y}, \sigma_{\text{NS}}] + \nu[e^{ik_z}, \sigma_{\text{NS}}] + \nu[e^{-i(k_x+k_y+k_z)}, \sigma_{\text{NS}}] \\ &= 0 + 0 + 0 + 1 = 1, \end{aligned} \tag{91}$$

$$\begin{aligned} \nu[q_B, \sigma_{\text{NS}}] &= \nu[1, \sigma_{\text{NS}}] + \nu[e^{i(k_x-k_y)}, \sigma_{\text{NS}}] + \nu[e^{i(k_y-k_z)}, \sigma_{\text{NS}}] + \nu[e^{i(k_z-k_x)}, \sigma_{\text{NS}}] \\ &= 0 + 0 + 0 + 0 = 0. \end{aligned} \tag{92}$$

Thus, the models $q_A$ and $q_B$ cannot be distinguished by the invariants $W_1$ and $w_2$, but they are detected by the $\nu$ invariant, which proves that there exists no adiabatic path connecting $q_A$ and $q_B$.

Equivalently, the direct sum $q_C = q_A \oplus q_B$ has trivial one-dimensional winding numbers $W_{1x} = W_{1y} = W_{1z} = 0$ and trivial second SW class $w_2(P_{q_C}) = 0$. Nevertheless, its $\nu$-invariant (independent of the spin structure) is $\nu[q_C, \sigma] = \nu[q_A, \sigma] + \nu[q_B, \sigma] = 1$, showing that $q_C$ cannot be adiabatically connected to the trivial $8 \times 8$ model $q_{\text{triv}} = \mathbf{1}_8$.

Moreover, by Lemma 3.4, one obtains an interesting constraint on the three-dimensional winding number: if a unitary matrix $h : T^3 \to U(4)$ satisfies

$$q_B h^* q_A^\dagger = h \tag{93}$$

at each point of $T^3$, then the three-dimensional winding number $W_3[h]$ must be odd.

## 4.3 Other models

We also summarize additional models that generate the $K$-group $KO^1(T^3) = \mathbb{Z}^{\oplus 3} \oplus \mathbb{Z}_2^{\oplus 4}$. Before doing so, we first confirm that if a symmetric unitary $q$ on the three-torus $T^3$ depends only on two momenta, then its $\nu$-invariant vanishes in the (NS,NS,NS) sector:

**Proposition 4.1.** *If $\boldsymbol{n} \cdot \nabla_{\boldsymbol{k}} q = 0$ for some $\boldsymbol{n} \in \mathbb{Z}^{\times 3}$, then $\nu[q, \sigma_{\mathrm{NS}}] = 0$.*

In fact, if $q$ depends only on $k_x, k_y$, we can split the Dirac operator into the $k_x, k_y$ and $k_z$ parts, $iD_A = iD_{\mathrm{2D}} + k_z \gamma^3$. The anticommutation relation $\{iD_{\mathrm{2D}}, \gamma^3\} = 0$ implies that the nonzero eigenvalues of $iD_{\mathrm{2D}}$ appear in $\pm\lambda$ pairs, so the eigenvalues of $iD_A$ appear as pairs $\pm\sqrt{\lambda^2 + k_z^2}$. Hence, only the zero modes of $iD_{\mathrm{2D}}$ can contribute to the $\xi$-invariant. These zero modes split into the $\gamma^3 = \pm 1$ sectors, each giving eigenvalues $\pm k_z$ for $iD_A$. Therefore, if the $k_z$ direction has NS boundary conditions, the spectrum cancels between positive and negative parts, leaving no contribution to the $\xi$-invariant.

As an explicit model with trivial $W_{1\mu}$ but nontrivial $w_2$, one can consider the $2 \times 2$ Dirac model

$$q_{2,\mu\nu} = (\cos k_\mu + \cos k_\nu - m)\mathbf{1}_2 + i(\sin k_\mu \sigma_x + \sin k_\nu \sigma_z), \quad 0 < m < 2, \tag{94}$$

where the behavior near $\boldsymbol{k} \simeq \boldsymbol{0}$ compactifies to a 2D sphere in the $k_\mu k_\nu$ plane, giving $w_2(P_{q_{2,\mu\nu}}) = dk_\mu dk_\nu$. Eq. (4.1) then Propositon 4.1 ensures $\nu[q_{2,\mu\nu}, \sigma_{\mathrm{NS}}] = 0$.

A model depending on all three momentum directions and compactifying to $S^3$ is the $4 \times 4$ Dirac model [9]:

$$q_3 = \cos k_x + \cos k_y + \cos k_z - m + i(\sin k_x \sigma_x + \sin k_y \sigma_y \tau_y + \sin k_z \sigma_z), \quad 1 < m < 3. \tag{95}$$

This model has trivial both $W_1$ and $w_2$, admits a global Takagi factorization $q = QQ^\top$, and its $\nu$-invariant is given by $\nu[q_3, \sigma_{\mathrm{NS}}] = \nu[\mathbf{1}_4, \sigma_{\mathrm{NS}}] + W_3[Q]$. Since $W_3[Q] \equiv 1 \bmod 2$ [9], the $\nu$-invariant is nontrivial.

The topological invariants of the models introduced in this section are summarized in the following table:

| | $W_{1x}$ | $W_{1y}$ | $W_{1z}$ | $w_{2,xy}$ | $w_{2,yz}$ | $w_{2,zx}$ | $\nu[q, \sigma_{\mathrm{NS}}]$ |
|---|---|---|---|---|---|---|---|
| $q_{1,(0,0,0)}$ | 0 | 0 | 0 | 0 | 0 | 0 | 0 |
| $q_{1,(1,0,0)}$ | 1 | 0 | 0 | 0 | 0 | 0 | 0 |
| $q_{1,(0,1,0)}$ | 0 | 1 | 0 | 0 | 0 | 0 | 0 |
| $q_{1,(0,0,1)}$ | 0 | 0 | 1 | 0 | 0 | 0 | 0 |
| $q_{1,(1,1,0)}$ | 1 | 1 | 0 | 0 | 0 | 0 | 0 |
| $q_{1,(0,1,1)}$ | 0 | 1 | 1 | 0 | 0 | 0 | 0 |
| $q_{1,(1,0,1)}$ | 1 | 0 | 1 | 0 | 0 | 0 | 0 |
| $q_{1,(1,1,1)}$ | 1 | 1 | 1 | 0 | 0 | 0 | 1 |
| $q_{2,xy}$ | 0 | 0 | 0 | 1 | 0 | 0 | 0 |
| $q_{2,yz}$ | 0 | 0 | 0 | 0 | 1 | 0 | 0 |
| $q_{2,zx}$ | 0 | 0 | 0 | 0 | 0 | 1 | 0 |
| $q_A$ | 0 | 0 | 0 | 1 | 1 | 1 | 1 |
| $q_B$ | 0 | 0 | 0 | 1 | 1 | 1 | 0 |
| $q_3$ | 0 | 0 | 0 | 0 | 0 | 0 | 1 |

$$\tag{96}$$

Here, the second SW class is expressed with coefficients as

$$w_2(P) = w_{2,xy}\, dk_x dk_y + w_{2,yz}\, dk_y dk_z + w_{2,zx}\, dk_z dk_x. \tag{97}$$

With this table, we confirm that, for instance, $q_A \oplus q_B$ is stably equivalent to $q_3$, i.e., there is an adiabatic path from $q_A \oplus q_B \oplus \mathbf{1}_n$ to $q_3 \oplus \mathbf{1}_{n+4}$ for some $n$.

## 5 Summary

In this paper, we have constructed the previously unresolved $\mathbb{Z}_2$ topological invariant for class CI band structures with PT and PC symmetry in three dimensions. Using the principal $O(n)$ bundle induced

by PT symmetry together with the real Berry connection, we defined the spin-CS action $\mathrm{CS}_{\mathrm{spin}}$ with $4\pi$ periodicity via the $\eta$-invariant. We further showed that PC symmetry quantizes this action to $\{0, 2\pi\}$, thereby establishing it as the $\mathbb{Z}_2$ invariant of class CI. The resulting invariant $\nu = \mathrm{CS}_{\mathrm{spin}}/(2\pi)$ is additive under direct sums of band structures, and in cases admitting a global Takagi factorization, it reduces to the previously known $\mathbb{Z}_2$ invariant [9].

An important remark is that the $\mathbb{Z}_2$ invariant $\nu$ generally depends on the spin structure. Since there is no canonical choice of spin structure, one must choose a spin structure by hand when using $\nu$ to characterize band structures.

With the $\nu$-invariant presented in this paper, the set of topological invariants for the eight real AZ classes defined by PT and PC symmetries is now complete in spatial dimensions up to three.

# Acknowledgements

We thank Jin Miyazawa, Hidenori Fukaya, Kantaro Ohmori, and Yuya Tanizaki for helpful discussions. This work was supported by JST CREST Grant No. JPMJCR19T2, and JSPS KAKENHI Grant Nos. JP22H05118 and JP23H01097.

# A    Some basics

Let $M$ be a closed oriented $d$-dimensional manifold, and let $H_{\boldsymbol{k}}$, $\boldsymbol{k} \in M$, be a gapped Hamiltonian defined on $M$, meaning that for every point $\boldsymbol{k} \in M$, the matrix $H_{\boldsymbol{k}}$ is Hermitian and has no zero eigenvalue. As notation, we write $\Phi_{\boldsymbol{k}} = (u_{1\boldsymbol{k}}, \ldots, u_{n\boldsymbol{k}})$ for the matrix whose columns are the eigenvectors with negative eigenvalues of $H_{\boldsymbol{k}}$, i.e., $H_{\boldsymbol{k}} u_{j\boldsymbol{k}} = E_{j\boldsymbol{k}} u_{j\boldsymbol{k}}$ with $E_{j\boldsymbol{k}} < 0$. We call $\Phi_{\boldsymbol{k}}$ the occupied state frame, where $n$ is the number of negative eigenvalues (the filling number). Introducing the flattened Hamiltonian $\mathrm{sgn}\, H_{\boldsymbol{k}} = \sum_j \mathrm{sgn}\,(E_{j\boldsymbol{k}}) u_{j\boldsymbol{k}} u_{j\boldsymbol{k}}^\dagger$, we can equivalently define $\mathrm{sgn}\, H_{\boldsymbol{k}} \Phi_{\boldsymbol{k}} = -\Phi_{\boldsymbol{k}}$. In general, $\Phi_{\boldsymbol{k}}$ cannot be chosen continuously on all of $M$; instead, continuity is guaranteed only locally on open subsets of $M$. One obtains the $U(n)$ Berry connection $A_{\boldsymbol{k}} = \Phi_{\boldsymbol{k}}^\dagger d\Phi_{\boldsymbol{k}}$, also defined locally. The Berry curvature is $F_{\boldsymbol{k}} = dA_{\boldsymbol{k}} + A_{\boldsymbol{k}}^2$. Under a gauge transformation $\Phi_{\boldsymbol{k}} \mapsto \Phi_{\boldsymbol{k}} W_{\boldsymbol{k}}$, the connection and curvature transform as $A_{\boldsymbol{k}} \mapsto W_{\boldsymbol{k}}^\dagger (A_{\boldsymbol{k}} + d) W_{\boldsymbol{k}}$ and $F_{\boldsymbol{k}} \mapsto W_{\boldsymbol{k}}^\dagger F_{\boldsymbol{k}} W_{\boldsymbol{k}}$, respectively.

## A.1    Chern numbers and Winding numbers

Let $d = 2m$ be even. Then the integer topological index, the Chern number, is defined as

$$ch_m[H_{\boldsymbol{k}}] := \frac{1}{m!} \left(\frac{i}{2\pi}\right)^m \int_M \mathrm{tr}\,[F_{\boldsymbol{k}}^m] \in \mathbb{Z}. \tag{98}$$

Let $d = 2m + 1$ be odd, and assume the chiral symmetry

$$\Gamma H_{\boldsymbol{k}} \Gamma^\dagger = -H_{\boldsymbol{k}}, \quad \Gamma^2 = 1. \tag{99}$$

Choosing a basis with $\Gamma = \sigma_z$, the flattened Hamiltonian takes the block form

$$\mathrm{sgn}\, H_{\boldsymbol{k}} = \begin{pmatrix} & q_{\boldsymbol{k}} \\ q_{\boldsymbol{k}}^\dagger & \end{pmatrix}, \quad q_{\boldsymbol{k}} \in U(n). \tag{100}$$

An integer winding number is then defined by

$$W_{2m+1}[q_{\boldsymbol{k}}] := \frac{m!}{(2m+1)!(2\pi i)^{m+1}} \int_M \mathrm{tr}\,\left[(q_{\boldsymbol{k}} dq_{\boldsymbol{k}}^{-1})^{2m+1}\right] \in \mathbb{Z}. \tag{101}$$

## A.2    CS action

Let $d = 3$ and $n$ be the number of occupied states. The Hamiltonian $H_{\boldsymbol{k}}$ determines a point in the classifying space $BU(n)$. Since the 3-dimensional cobordism group vanishes, $\Omega_3^{\mathrm{SO}}(BU(n)) = 0$, one can

extend $H_{\boldsymbol{k}}$ to a 4-dimensional manifold while keeping the gap open. We denote this extension by $\tilde{H}_{\boldsymbol{k}}$, with $\boldsymbol{k} \in X$, $\partial X = M$. Using the Berry curvature $\tilde{F}_{\boldsymbol{k}}$ of $\tilde{H}_{\boldsymbol{k}}$, the Chern-Simons action is defined as

$$\mathrm{CS}_3(A_{\boldsymbol{k}}) := -\frac{1}{4\pi} \int_X \mathrm{tr}\,[\tilde{F}_{\boldsymbol{k}}^2] \quad \mathrm{mod}\ 2\pi. \tag{102}$$

The mod $2\pi$ ambiguity arises as follows: if $\tilde{H}'_{\boldsymbol{k}}$, $\boldsymbol{k} \in X'$, is another extension, the ambiguity is given by the second Chern number on $X \cup (-X')$,

$$ch_2 = -\frac{1}{8\pi^2} \int_{X \cup (-X')} \mathrm{tr}\,[\tilde{F}_{\boldsymbol{k}}^2], \tag{103}$$

which takes integer values. Under a smooth global gauge transformation $\Phi_{\boldsymbol{k}} \mapsto \Phi_{\boldsymbol{k}} W_{\boldsymbol{k}}$ on $M$, the Chern-Simons action transforms as

$$\mathrm{CS}_3(A_{\boldsymbol{k}}) \mapsto \mathrm{CS}_3(A_{\boldsymbol{k}}) + 2\pi W_3[W_{\boldsymbol{k}}]. \tag{104}$$

If a global occupied-state frame exists on $M$, one can write

$$\mathrm{CS}_3(A_{\boldsymbol{k}}) = -\frac{1}{4\pi} \int_M \mathrm{tr}\,\left(A_{\boldsymbol{k}} dA_{\boldsymbol{k}} + \tfrac{2}{3} A_{\boldsymbol{k}}^3\right). \tag{105}$$

If the Hamiltonian $H_{\boldsymbol{k}}$ has additional symmetries, these restrict the possible values of CS. In the presence of the chiral symmetry (99), taking $\Gamma = \sigma_z$, the block form (100) defines $q_{\boldsymbol{k}}$, and one obtains a global frame $\Phi_{\boldsymbol{k}} = \frac{1}{\sqrt{2}}(q_{\boldsymbol{k}}, -\mathbf{1}_n)^\top$. Then the Berry connection has the global form $A_{\boldsymbol{k}} = \frac{1}{2} q_{\boldsymbol{k}}^\dagger dq_{\boldsymbol{k}}$ and it follows that

$$\mathrm{CS}_3(A_{\boldsymbol{k}}) \equiv \pi W_3[q_{\boldsymbol{k}}] \pmod{2\pi}. \tag{106}$$

Thus, in the presence of chiral symmetry, the CS action is quantized to $\mathbb{Z}_2$ and equals the parity of the 3-dimensional winding number [5].

# B    Invariants for AZ class with PT and PC symmetries

Since chiral symmetry is equivalent to an internal symmetry, we focus here only on invariants in the real AZ classes. Let $\boldsymbol{k} \in T^d$ denote the momentum in $d$-dimensional space, and let $H_{\boldsymbol{k}}$ be a Hamiltonian that remains gapped at $E = 0$ throughout $T^d$. The relevant symmetries are PT symmetry

$$TH_{\boldsymbol{k}}^* T^\dagger = H_{\boldsymbol{k}}, \quad TT^* = \pm 1, \tag{107}$$

and/or PC symmetry

$$CH_{\boldsymbol{k}}^* C^\dagger = -H_{\boldsymbol{k}}, \quad CC^* = \pm 1, \tag{108}$$

whose combinations generate the real AZ classes. The possible combinations of $T$ and/or $C$ and corresponding eight AZ classes and topological classification are summarized in Table 1.

In Table 1, $\mathbb{Z}$ invariants coincide with Chern numbers and winding numbers defined without imposing PT or PC symmetries. Therefore, in what follows, we summarize only the explicit constructions of the $\mathbb{Z}_2$ invariants.

## B.1    AI

Without loss of generality, we may take $T = 1$. Let $\{U_\alpha\}_\alpha$ be a good covering of the parameter space $X$. On each patch, the local frame can be chosen real in momentum space, i.e. $(\Phi_{\boldsymbol{k}}^\alpha)^* = \Phi_{\boldsymbol{k}}^\alpha$. On the overlaps $\boldsymbol{k} \in U_\alpha \cap U_\beta$, the relation $\Phi_{\boldsymbol{k}}^\beta = \Phi_{\boldsymbol{k}}^\alpha S_{\boldsymbol{k}}^{\alpha\beta}$ defines transition functions $S_{\boldsymbol{k}}^{\alpha\beta} \in O(n)$. They satisfy the cocycle condition $S_{\boldsymbol{k}}^{\alpha\beta} S_{\boldsymbol{k}}^{\beta\gamma} = S_{\boldsymbol{k}}^{\alpha\gamma}$ on triple overlaps $\boldsymbol{k} \in U_\alpha \cap U_\beta \cap U_\gamma$. Thus, a class AI Hamiltonian defines a principal $O(n)$ bundle $P$ over the parameter space $X$, in particular its SW classes $w_i(P) \in H^i(X, \mathbb{Z}_2)$. For $d = 1$, the $\mathbb{Z}_2$ invariant is the first SW class, which coincides with the Berry phase quantized to $\{0, \pi\}$. For $d = 2$, the $\mathbb{Z}_2$ invariant is the second SW class.

Table 1: Classification of real AZ classes arising from PT and PC symmetries.

| AZ class | $TT^*$ | $CC^*$ | $d = 0$ | $d = 1$ | $d = 2$ | $d = 3$ |
|---|---|---|---|---|---|---|
| AI | 1 | 0 | $\mathbb{Z}$ | $\mathbb{Z}_2$ | $\mathbb{Z}_2$ | 0 |
| BDI | 0 | 1 | $\mathbb{Z}_2$ | $\mathbb{Z}_2$ | 0 | $2\mathbb{Z}$ |
| D | 0 | 1 | $\mathbb{Z}_2$ | 0 | $2\mathbb{Z}$ | 0 |
| DIII | $-1$ | 1 | 0 | $2\mathbb{Z}$ | 0 | 0 |
| AII | $-1$ | 0 | $2\mathbb{Z}$ | 0 | 0 | 0 |
| CII | $-1$ | $-1$ | 0 | 0 | 0 | $\mathbb{Z}$ |
| C | 0 | $-1$ | 0 | 0 | $\mathbb{Z}$ | $\mathbb{Z}_2$ |
| CI | 1 | $-1$ | 0 | $\mathbb{Z}$ | $\mathbb{Z}_2$ | $\mathbb{Z}_2$ |

## B.2   BDI

Taking $T = 1$ and $C = \sigma_z$, the flattened Hamiltonian has the form

$$\operatorname{sgn} H_{\boldsymbol{k}} = \begin{pmatrix} & q_{\boldsymbol{k}} \\ q_{\boldsymbol{k}}^\dagger & \end{pmatrix}, \quad q_{\boldsymbol{k}} \in O(n). \tag{109}$$

For $d = 0$, the $\mathbb{Z}_2$ invariant is given by $\det q_{\boldsymbol{k}} \in \{\pm 1\}$. For $d = 1$, the $\mathbb{Z}_2$ invariant corresponds to $\pi_1[O(n)] = \mathbb{Z}_2$. One way to construct it concretely is as follows: for any nontrivial loop $\ell : S^1 \to X$ in the parameter space, discretize the loop into sufficiently fine points $\ell_1, \dots, \ell_N, \ell_{N+1} = \ell_1$. Compute the relative orthogonal matrices $\delta q_j := q_{\ell_{j+1}} q_{\ell_j}^\top$, lift each $\delta q_j$ numerically to $\widetilde{\delta q}_j \in \mathrm{Pin}_+(n)$, and take the product $\widetilde{\delta q}_N \cdots \widetilde{\delta q}_1 = \pm \mathbf{1}$. The overall sign $\pm 1$ gives the $\mathbb{Z}_2$ invariant.

## B.3   D

For $C = 1$, the symmetry condition reads $H_{\boldsymbol{k}}^\top = -H_{\boldsymbol{k}}$. For $d = 0$, the $\mathbb{Z}_2$ invariant is given by the sign of the Pfaffian, $\operatorname{sgn} \operatorname{Pf}(H_{\boldsymbol{k}}) \in \{\pm 1\}$.

## B.4   C

For $d = 3$, the $\mathbb{Z}_2$ invariant is the CS action quantized to $\{0, \pi\}$. This can be shown using dimensional reduction [24]. The symmetry condition is $CH_{\boldsymbol{k}}^* C^\dagger = -H_{\boldsymbol{k}}$, with $CC^* = -1$. Ignoring this symmetry, one can extend the Hamiltonian into a fourth dimension so that $\mathrm{CS}_3(A_{\boldsymbol{k}})$ is defined. Let $\theta \in [0, \pi]$ parametrize this extension, and denote the extended Hamiltonian by $\tilde{H}_{\boldsymbol{k}, \theta}$. By $PC$ symmetry, define the extension in the $-\theta$ direction as $\tilde{H}_{\boldsymbol{k}, -\theta} = C\tilde{H}_{\boldsymbol{k}, \theta}^* C$. Then the contributions to the second Chern number $ch_2[\tilde{H}_{\boldsymbol{k}, \theta}]$ from $(\boldsymbol{k}, \theta)$ and $(\boldsymbol{k}, -\theta)$ agree[10]. Hence,

$$\mathrm{CS}_3(A_{\boldsymbol{k}}) \equiv \pi \, ch_2[\tilde{H}_{\boldsymbol{k}}] \mod 2\pi, \tag{110}$$

which shows that it is quantized to $\mathbb{Z}_2$ values.

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
