# Peer review of "$\mathbb{Z}_2$ topological invariant in three-dimensional PT- and PC-symmetric class CI band structures"

_SciPost Physics Core_

## Round 2 · Referee Report · Anonymous (Referee 1) · 2025-10-21

Report

In this work, the author studies the 3D Z_2 topological invariant for the PT and CP symmetric class CI valence bands. At each k, the Bloch Hamiltonian is characterized by a symmetric unitary matrix q(k). Previously the Takagi factorization q(k)=Q(k)Q(k)^T had been employed to formulate the topological invariant, where a global Takagi factor Q(k) was assumed over the Brillouin torus and the Z2 nature comes from the O(N) gauge freedom of the Takagi factorization.

In this work, the author points out that in general the Takagi factorization gives rise to an emergent principal O(N)-bundle, whose topological class can be characterized by its Stiefel-Whitney classes w1 and w2, and the global Takagi factor Q(k) exists if and only if the principal O(N)-bundle is trivial, namely w1=w2=0. In addition, w1 and w2 are just the Stiefel-Whiteney classes as lower dimensional topological invariants protected by PT symmetry. Thus, a global Takagi factor Q(k) exists if and only if lower-dimensional topological invariants w1 and w2 are all trivial.

Based on these observations, the author formulated a general 3D topological invariant, namely the spin Chern Simons term of the principal O(N)-bundle with the canonical connection (Eq. 13) from the Takagi factorization. The definition of the spin Chern Simons term requires a presumed spin structure, and the term depends on the spin structure in the presence of nontrivial w1 and w2.

While this is certainly an important contribution to this field, I have a concern about this spin-structure dependency. Physically q has trivial or nontrivial 3D topological invariant should be independent of the spin structure \sigma, since \sigma is just a convention. I elaborate this point from two aspects as follows.

First, when the lower dimensional topological invariants are trivial, there exists q_0(k) and q_1(k) with trivial and nontrivial 3D topological invariants, respectively, which can be determined by global Takagi factors. Then, the 3D Z_2 invariant \nu(q_1\oplus q, \sigma) should be 0 if q is nontrivial and should be 1 if q is trivial. This specifies a physically preferred spin structure.

Second, the splitting form of the K group in Eq. 16 implies that there exists a 3D Z2 invariant which is independent of the lower dimensional topological invariants. But now, the spin Chern-Simons term depends on lower dimensional topological invariants from its dependence on the spin structure.

Below, some additional comments are listed for the author’s consideration.
1. I wonder whether w_2(P_q) of the emergent O(N)-bundle is the same as the second Stiefel Whitney class of the real valence bands from PT symmetry.
2. For 3-torus, the eight spin structures correspond to the eight possible periodic/anti-periodic boundary conditions, which may be introduced in the manuscript for readers who are not familiar with the abstract concept of spin structure.
3. In the paragraph below Eq. 11, the gauge transformation of A misses the transposition of the first S.
4. In Eq. 13, Hermitian conjugation is missed for the first Q.
5. The connection in Eq. 13 and that in Eq. 15 are related by a unitary gauge transformation, which may be explicitly stated.

Recommendation

Ask for minor revision

---

## Editorial Decision

in_refereeing